# The Influence of Surgical Experience and Bone Density on the Accuracy of Static Computer-Assisted Implant Surgery in Edentulous Jaws Using a Mucosa-Supported Surgical Template with a Half-Guided Implant Placement Protocol—A Randomized Clinical Study

**DOI:** 10.3390/ma13245759

**Published:** 2020-12-17

**Authors:** Márton Kivovics, Dorottya Pénzes, Orsolya Németh, Eitan Mijiritsky

**Affiliations:** 1Department of Community Dentistry, Semmelweis University, 1088 Budapest, Hungary; penzes.dorottya@dent.semmelweis-univ.hu (D.P.); nemeth.orsolya@dent.semmelweis-univ.hu (O.N.); 2Head and Neck Maxillofacial Surgery, Department of Otolaryngology, Tel-Aviv Sourasky Medical Center, Sackler Faculty of Medicine, Tel-Aviv University, Tel Aviv-Yafo 6997801, Israel; mijiritsky@bezeqint.net; 3The Maurice and Gabriela Goldschleger School of Dental Medicine, Sackler Faculty of Medicine, Tel-Aviv University, Tel-Aviv 6997801, Israel

**Keywords:** dental implant, surgical guide, static navigation, computer-assisted implant surgery, accuracy, surgical experience, bone density, grey level measurements, Hounsfield Unit

## Abstract

The aim of our randomized clinical study was to analyze the influence of surgical experience and bone density on the accuracy of static computer-assisted implant surgery (CAIS) in edentulous jaws using a mucosa-supported surgical template with a half-guided implant placement protocol. Altogether, 40 dental implants were placed in the edentulous jaws of 13 patients (novice surgeons: 18 implants, 6 patients (4 male), age 71 ± 10.1 years; experienced surgeons: 22 implants, 7 patients (4 male), age 69.2 ± 4.55 years). Angular deviation, coronal and apical global deviation and grey level measurements were calculated for all implants by a blinded investigator using coDiagnostiX software. 3DSlicer software was applied to calculate the bone volume fraction (BV/TV) for each site of implant placement. There were no statistically significant differences between the two study groups in either of the primary outcome variables. There was a statistically significant negative correlation between angular deviation and both grey level measurements (R-value: −0.331, *p* < 0.05) and BV/TV (R-value: −0.377, *p* < 0.05). The results of the study suggest that surgical experience did not influence the accuracy of implant placement. The higher the bone density at the sites of implant placement, the higher the accuracy of static CAIS.

## 1. Introduction

In contemporary implantology, a multidisciplinary approach to oral rehabilitation requires backwards planning. The number, dimensions, position and inclination of dental implants placed are determined by prosthetic planning, which facilitates optimal esthetic results, optimal function, ideal biomechanical loading for all components of the prosthesis and long-term stability of soft and hard tissues surrounding the implants [1,2,3]. Surface treatment of dental implants and fixtures during production plays an important role in muco-integration. Treatment modalities for peri-implantitis should be chosen with regard to the fact that they may alter surface physicochemical properties and influence the health of soft tissues surrounding the implants [4,5]. Computer-assisted design and computer-assisted manufacturing (CAD/CAM) enable virtual planning of the prosthesis and position of the dental implants, which allows better communication between the prosthodontist, the implant surgeon and the dental technician. Virtual planning promotes the management of patient expectations by way of visualization of the expected treatment outcomes [3,6].

During static computer-assisted implant surgery (CAIS), the planned implant positions are reproduced by a surgical template. In a full-guided procedure, implant bed preparation and placement are both guided by the template, whereas in a half-guided approach, only the initial steps of the implant bed preparation are guided by the template; finalization of the implant beds and implant placement is carried out in a free-handed fashion. Both approaches reduce surgical time and, therefore, postoperative discomfort [2,7]. The full-guided approach enables flapless surgery, which further reduces postoperative morbidity [7,8,9,10]. However, flap elevation and a half-guided approach enhance cooling of the drills used, promote the preservation of keratinized mucosa and allow slight correction of implant positions if deemed necessary [1,3,11,12].

According to the literature, in the case of static CAIS, mean deviations of 0.9 mm at the coronal level, 1.3 mm at the apex and 3.5° angular deviation from the planned implant positions are observed [13]. According to the results of randomized clinical trials, full-guided CAIS enables more accurate implant placement than the half-guided approach [14,15,16,17]. The inaccuracy of CAIS is commensurable with the desired bone width of at least 1.5 mm on the oral and vestibular aspects of the implant, which is necessary for the long-term stability of the hard tissues around the implant [3].

This overall error is a cumulative sum of errors from imaging, transformation of the virtual plan to a surgical template, fabrication of the guide, positioning of the template during surgery and errors during surgery itself [1,2,13]. According to the literature, implant placement may be carried out more accurately with tooth-supported surgical templates than with mucosa-supported templates. Implant placement with the use of bone-supported templates is the least precise of the three [1,3,7,13].

According to the findings of the 4th European Association for Osseointegration (EAO) Consensus Conference 2015, there is no difference between the accuracy of implant placement in the maxilla and the mandible [13].

A factor which might affect the accuracy of static guided implant placement is the experience of the surgeon. The role of experience has been investigated via in vitro studies [18,19,20,21,22,23]. However, few clinical studies are available on the subject [24,25,26]. A systematic review by Cheongbeom et al. concluded that further randomized controlled clinical studies are required to determine whether there is a difference in the accuracy of guided implant placement depending on which jawbone receives the implants and whether surgical experience influences the accuracy [27].

There is limited evidence in the literature on the influence of recipient bone quality on the accuracy of static CAIS. Few cadaver [28] and clinical [29,30,31,32,33] studies have been conducted with differing methodology and results to correlate cortical bone thickness, bone quality and bone density with the accuracy of guided implant placement.

The aim of our randomized controlled clinical study was to analyze the influence of surgical experience on the accuracy of static CAIS in the edentulous jaws using a mucosa-supported surgical template stabilized by pins with a half-guided implant placement protocol. The secondary aim of this study was to analyze the influence of bone density on the accuracy of guided implant placement. Because of the controversy in the literature regarding the role of experience and bone density on the accuracy of CAIS, our null hypotheses were that there is no difference in the accuracy of CAIS performed by experienced and novice implant surgeons and that there is no correlation between bone density and the accuracy of CAIS.

## 2. Materials and Methods

### 2.1. Study Design

A randomized clinical study was designed to address the aim of the study. The study was approved by Semmelweis University’s Regional, Institutional Scientific and Research Ethics Committee (109/2020) and it was conducted in accordance with the Helsinki declaration. Surgical procedures carried out during the study were thoroughly explained to the patients enrolled. Patients signed informed consent documents.

The patient sample size was chosen based on previous in vitro and clinical studies [18,24,34]. Cushen et al. found that the angular deviation of implant placement was 2.60° ± 1.25° in a study group where surgical interventions were carried out by experienced surgeons and 3.96° ± 1.64° in a study group where surgery was carried out by novice implant surgeons [18]. According to the result of sample size calculation, if α (false positive rate) was determined at 0.05, the number of cases required was 18–22 to achieve a power of 90–95%. According to the results of Marei et al., the buccolingual deviations of implants placed with CAIS were 3.7 ± 3.35 mm and 8.5 ± 6.3 mm in experienced and novice groups, respectively [24]. According to the result of sample size calculation, if α (false positive rate) was determined at 0.05, the required number of cases was 12–14 to achieve a power of 90–95%. Consequently, in the present study, approximately 20 cases (number of implants placed) were enrolled in each study group.

Patients of the Department of Community Dentistry, Semmelweis University, presenting with an edentulous lower and/or upper jaw, who were more than 18 years of age and who needed implant-supported restorations were included in the present study. The anatomic inclusion criteria were clinically and radiologically healthy alveolar ridges with a horizontal dimension of at least 7 mm and a vertical dimension of at least 10 mm from the vital anatomical landmarks. A healing period of at least 3 months following the last extraction in the proximity of planned implant placement was observed in all cases.

The exclusion criteria were as follows:History of systemic diseases (i.e., osteoporosis, diabetes mellitus) or medications (bisphosphonates, receptor activator of nuclear factor kappa-Β (RANK) ligand inhibitor monoclonal antibodies, corticosteroids, etc.) known to alter bone metabolism;History of tumors or irradiation therapy in the head and neck region;History of uncontrolled medical or psychiatric disorders;Unwillingness to return for follow-up appointments;Pregnancy;Smoking;Inability to perform proper oral hygiene.

Simple randomization was carried out using the random number generator function of Excel (Microsoft, Redmond, WA, USA) to determine whether patients would be included in the test or control group. In the test group, static CAIS was carried out by three novice surgeons who had completed theoretical training in implant dentistry, had placed less than 20 dental implants and had carried out only free-hand implant placements previous to the study, while in the control group, guided implant placement was carried out by two experienced surgeons who had placed more than 100 implants over the course of the year before the study autonomously using both conventional and guided approaches.

In the present study, the dependent primary outcome variables dealing with the accuracy of implant placement were angular deviation, coronal global deviation and apical global deviation. The independent variables were the level of surgical experience, grey level measurements in Hounsfield Units (HU) and bone volume fraction (BV/TV), the relative volume of calcified tissue in the selected volume of interest.

### 2.2. Preoperative and Postoperative Imaging

Records of the soft tissues were captured by optical scan of the study cast. Another scan of the study cast was carried out with five gutta-percha markers (Diadent Group International, Chungcheongbuk-do, Korea) fixed on the alveolar ridge of the study cast. Bite registration was fabricated with the gutta-percha markers incorporated in wax and an acrylic base. The centric occlusion position of the jaws was recorded with the bite registration and was reproduced during preoperative cone beam computed tomography (CBCT) scanning. Finally, a third optical scan was carried out with the diagnostic wax-up placed on the study cast. Optical scanning was carried out using a dental laboratory desktop scanner (3Series, Dental Wings, Montreal, CA, USA), resulting in the three Standard Tessellation Language (STL) files later applied for planning the surgical guide.

CBCT imaging (PaX-Reve3D, Vatech, Hwaseong, Korea) was carried out prior to guided dental implant placement in the planning stage with the bite registration in place (preoperative CBCT) and again 6 months after dental implant placement (postoperative CBCT). The scanning conditions were constant at 250 µm isotropic voxel size with 360° rotation, 89 kV tube voltage, 4.9 mA tube current and 24 s exposure time for all specimens with a 15 × 15 cm field of view (FOV).

### 2.3. Preoperative Planning

All surgical guides used in the present study were designed by the same surgeon (M.K.), who supervised all surgical procedures in both study groups. Surgical planning was carried out using coDiagnostiX software, version 10.2 (Dental Wings, Montreal, CA, USA). The STL file of the study cast with the gutta-percha markers in place was registered with the Digital Imaging and Communications in Medicine (DICOM) data of the preoperative CBCT reconstruction by surface registration. The STL file of the study cast of the edentulous jaw without the gutta-percha markers and the study cast with the diagnostic wax-up were registered with the STL file of the study cast with the gutta-percha markers in place. In the present study, implant-borne overdentures anchored and supported by 2–4 implants or fixed prostheses supported by 4–6 implants were planned on the edentulous jaws. Prosthetically ideal implant positions were determined and a surgical guide for half-guided implant placement was designed with 2-mm-diameter sleeves (Article number HN001, Hager & Meisinger GmbH, Neuss, Germany) to guide the pilot drill (Article number HN011, Hager & Meisinger GmbH, Neuss, Germany) and 3 template fixation pins (Straumann Template Fixation Pin, Article number 034.282, Straumann GMBH, Basel, Switzerland) were planned to keep the surgical guide in place. All surgical guides were manufactured by stereolithography and sterilized prior to surgery. Figure 1 presents the planning of one of the cases.

### 2.4. Surgical Procedure

Patients rinsed with 0.2% chlorhexidine solution for 1 min before surgery. Under local anesthesia, the surgical guide was fixed using template fixation pins (Straumann Template Fixation Pin, Article number 034.282, Straumann GMBH, Basel, Switzerland). Pilot osteotomies were carried out using a pilot drill 2 mm in diameter (Article number HN011, Hager & Meisinger GmbH, Neuss, Germany). The fixation pins and the surgical guide were removed and a full-thickness flap was elevated from a crestal incision. Implant osteotomies were finalized according to the implant manufacturer’s instructions. In the case of 3.8-mm-diameter implants, a single 3.5-mm-diameter tapered drill specific to the length of the implant was used. In the case of 4.3 mm implants, in addition to the 3.5-mm-diameter drill, a 4.1-mm-diameter tapered drill specific to the length of the implant was used to finalize the osteotomies, with external cooling at a drill rotation speed of 800 rpm. Additional bone tapping was carried out in dense bone. Dental implants (Denti Root Form Plus, Denti-Systems Ltd., Szentes, Hungary) were placed non-submerged. Wound margins were stabilized with single interrupted sutures. Antibiotics (1 g amoxicillin-clavulanate twice a day for 5 days or, in case of side effects or known allergy to penicillin, 300 mg clindamycin 4 times a day for 4 days), non-steroid anti-inflammatory drugs (50 mg diclofenac 3 times a day for 3 days) and chlorhexidine mouthwash (twice a day for 14 days) were prescribed. Sutures were removed after 7 days. The base of preexisting dentures was modified to accommodate the healing abutments during the 3-month healing period. Figure 2 presents the surgical intervention.

### 2.5. Data Acquisition

Using coDiagnostiX software, version 10.2 (Dental Wings, Montreal, CA, USA), the DICOM data of the pre- and postoperative CBCT reconstructions were registered and the angular deviation, coronal global deviation and apical global deviation were calculated for all implants using the treatment evaluation plug-in of the software by an investigator (D.P.) blinded to the experience level of the surgeon. Grey level measurements for every implant were calculated by the coDiagnostiX software in HU based on the planned implant positions. Figure 3 presents the measurement of the primary outcome variables in the Treatment Evaluation plug-in of the planning software.

To assess the bone volume at the sites of implant placement, the width and height of the alveolar ridge were measured in the coDiagnostiX software. The width of the ridges was measured at the platform level of the implants planned; the ridge height was measured from the center of the implant platform along the axis of the planned implant position to the closest anatomical landmark.

DICOM data of the pre- and postoperative CBCT reconstructions were registered by landmark registration using 3DSlicer 4.10.2 software (The Brigham and Women’s Hospital, Inc., Boston, MA, USA). The segment editor module was used to manually determine the volume of interest (VOI)—the volume of bone on the preoperative segmentation—which received the implant by changing the opacity of the layers of the pre- and postoperative CBCT. The BoneTexture Extension was used to calculate BV/TV for each site of implant placement. Figure 4 presents the calculation of BV/TV using the BoneTexture Extension of 3DSlicer 4.10.2 (The Brigham and Women’s Hospital, Inc., Boston, MA, USA).

### 2.6. Statistical Analysis

A Shapiro–Wilk’s test (*p* < 0.05) and visual inspection of the histograms, normal Q–Q plots and box plots were carried out to determine whether the angular deviation, coronal global deviation and apical global deviation of the dental implants placed by novice and experienced surgeons were distributed approximately normally. It was concluded that the coronal and apical global deviation values for both experience levels showed a normal distribution, while the angular deviation values for both experience levels did not. One-way ANOVA test was carried out to compare the coronal and apical global deviation of implants placed by experienced and novice surgeons and Mann–Whitney’s U test was carried out to compare the angular deviation data of implants placed by the two groups.

Spearman’s test (two-tailed) was carried out to correlate the angular deviation and coronal and apical global deviation data to the BV/TV and grey level data.

Values of *p* < 0.05 were considered statistically significant.

## 3. Results

### 3.1. The Influence of Surgical Experience on the Accuracy of CAIS

Altogether, 40 dental implants were placed in the edentulous jaws of 13 patients (test group, novice surgeons: 18 implants placed in 6 patients (4 male), age 71 ± 10.1 years; control group, experienced surgeons: 22 implants placed in 7 patients (4 male), age 69.2 ± 4.55 years). The grey level was 613.4 ± 304.7 HU (663.7 ± 294.5 HU and 576.8 ± 313.5 HU for novice and experienced groups, respectively) and BV/TV was 0.6761 ± 0.3644 (0.7294 ± 0.3392 and 0.6373 ± 0.3847 for novice and experienced groups, respectively) calculated at the sites of implant placement. The width of the alveolar ridge at the sites of implant placement was 7.688 ± 1.094 mm (7.567 ± 0.7300 mm and 7.786 ± 1.329 mm for novice and experienced groups, respectively). The height of the alveolar ridge at the sites of implant placement was 15.89 ± 4.011 mm (16.02 ± 4.624 mm and 15.78 ± 3.542 mm for novice and experienced groups, respectively).

There were no statistically significant differences observed between the two groups in either of the primary outcome variables. The angular deviations of the implants placed were 6.544 ± 5.393 mm and 7.177 ± 4.214 mm for the test and control groups, respectively. The coronal global deviation was 1.987 ± 0.7049 mm for the novice group and 1.879 ± 0.7893 mm for the experienced group. The apical global deviations were 1.954 ± 0.6853 mm and 2.124 ± 0.8373 mm for the test and control groups, respectively. Table 1 presents the descriptive and comparative statistics of the primary outcome variables between the two study groups.

### 3.2. The Influence of Bone Density on the Accuracy of CAIS

There was a statistically significant negative correlation between the angular deviation of the implants and both grey level measurements (R-value: −0.331, *p* < 0.05) and BV/TV (R-value: −0.377, *p* < 0.05). The correlation was not statistically significant between either of the global deviation datasets and the grey level measurements or BV/TV. Table 2 presents the correlation statistics between the datasets dealing with the accuracy of implant placement and the bone density datasets.

## 4. Discussion

According to the literature, mucosa-supported templates are more accurate than bone-supported templates. Another approach more accurate than mucosa-supported templates that may be used in case of an edentulous jaw is an implant-supported template. However, this requires an additional surgical procedure to insert temporary implants, which increases patient discomfort and costs [1,2,3,8,14]. Both full-guided and half-guided static CAIS significantly reduce the length of surgery, which decreases postoperative morbidity and ensures that the implant positions achieved are closer to the prosthetically ideal, planned implant positions compared to those achieved by free-hand surgery [1,2,8,14,35]. Full-guided surgery further decreases the length of intervention by guiding every step of implant placement and further decreases postoperative morbidity by promoting flapless surgery [7,8,9,10,14]. However, during half-guided surgery, the template is removed for the finalization of the implant bed and does not inhibit the cooling of the drills [11,12]. Furthermore, half-guided surgery provides the possibility of raising a flap which allows the surgeon to correct an implant position in cases of inaccuracies and preserve the keratinized mucosa, which would be sacrificed by tissue punches in the case of a full-guided approach [14]. These advantages of the half-guided approach are relevant in those edentulous jaws where alveolar atrophy combined with the narrowing of keratinized mucosa may be advanced [1,2,3,8,14]. That is why, in the present study, half-guided CAIS with a mucosa-supported template fixed by pins was applied to navigate implant surgery, because this approach has some advantages over both free-hand and full-guided implant placement. We believe that these characteristics make the half-guided mucosa-supported template ideal for the training of novice implant surgeons.

In the literature, there is controversy over the question of whether surgical experience affects the accuracy of static CAIS. Several in vitro studies have suggested that experienced surgeons achieve more accurate implant placement using guided surgery in both partially [21,22] and fully edentulous jaws [18], using both full-guided [18,21,22] and half-guided [21] approaches. However, some studies have failed to support this [20,23].

Few randomized clinical studies have been published on the role of surgical experience in the accuracy of CAIS. In a cohort study by Van de Wile et al., implants were placed in edentulous jaws via a full-guided approach using a mucosa-supported template anchored by pins. According to the results of this study, there was no statistically significant difference in the accuracy of implant placement between novice and expert surgeons [25]. In a randomized clinical study by Marei et al., surgeons of different experience levels performed half-guided implant placement on partially edentulous jaws using a tooth-supported template. The results of their study suggest that experienced surgeons achieve more accurate implant positions compared to novice surgeons [24]. In a pilot study by Cassetta et al., an inexperienced group and an experienced group of surgeons performed static full-guided CAIS using a bone-supported template anchored by pins. No statistically significant differences were found in the accuracy of implant placement between the two groups [26].

In the present study, there was no significant difference in the primary outcome variables (angular deviation, coronal global deviation and apical global deviation of the implants placed) between the two groups, which supported our null hypothesis; surgical experience did not seem to influence the accuracy of half-guided static CAIS with a mucosa-supported template anchored by pins in the edentulous jaw.

The influence of bone density on the accuracy of static CAIS is also controversial, with some studies reporting a statistically significant negative correlation between bone density and the accuracy of CAIS [31,32]. However, according to other studies, no correlation [28,33] or statistically significant positive correlation has been observed between bone density and the accuracy of CAIS [29].

In their clinical study, Ochi et al. carried out full-guided surgery with mucosa-supported, pin-anchored templates on edentulous jaws and concluded that implants tend to be placed deeper than planned at sites with lower bone density and more superficially at sites with high bone density [30]. In a study by Putra et al., half-guided implant placement was carried out with a tooth-supported template and the results suggested a weak, statistically significant negative correlation between bone density and the accuracy of implant placement [31]. Ozan et al. found in their study that there was a statistically significant negative correlation between bone density and the accuracy of guided implant placement when performing half-guided static CAIS with mucosa-supported templates [32].

In their clinical study, Jones at al. could not find a statistically significant correlation between bone density and the accuracy of CAIS [33]. According to the results of a cadaver study by Noharet et al., no statistically significant correlation was found between the accuracy of guided implant placement and bone density [28].

Cassetta et al. performed static full-guided CAIS in edentulous and partially edentulous jaws using tooth- and mucosa- or mucosa-supported templates held in place manually or using template fixation pins. Contrary to other studies, according to their results, there was a positive correlation between angular deviation and bone density and the authors concluded that the greater the bone density at an implant site, the greater the inaccuracy of guided implant placement [29].

Most studies have employed conventional CT scans for preoperative planning and postoperative control [28,29,31,32]. However, other studies, such as the present one, have used CBCT for planning implant placement, designing templates and conducting follow-up [30,33]. Bone density measurements derived from CBCT reconstructions are inaccurate and should be generally avoided because of the artefacts and variable scanning conditions [36,37,38]. The literature suggests that micro-morphometric measurements based on CBCT data may be a reliable predictor of bone quality [39,40,41,42,43,44,45,46]. Consequently, in the present study, in addition to the grey level measurements of the CAIS software, bone density was determined by BV/TV measurements on the preoperative CBCT. The sites of implant placement were determined by registering pre- and postoperative CBCT data.

According to the results of the present study, there is a weak, statistically significant negative correlation between bone density (presented in both grey level measurements and bone volume fraction) and angular deviation. Our null hypothesis was rejected. The higher the bone density at the sites of implant placement, the higher the accuracy of static CAIS in edentulous jaws with a mucosa-supported half-guided template anchored by pins. In clinical practice, numerous factors besides bone density may influence the accuracy of CAIS. Because of the weak correlation between bone density and the accuracy of CAIS reported in the literature [30,31] and in the present study, to control these risk factors, further in vitro studies may be required to assess the role of bone density in the accuracy of CAIS. A possible interpretation of this result might be that it is more difficult to deviate from the path of the pilot drill in denser bone.

## 5. Conclusions

In this randomized clinical study, half-guided static CAIS with a mucosa-supported, pin-anchored template was carried out to place implants in edentulous jaws. The results of the study show that the experience of the surgeon did not influence the accuracy of implant placement. The higher the bone density at the sites of implant placement, the higher the accuracy of static CAIS.

## Figures and Tables

**Figure 1 materials-13-05759-f001:**
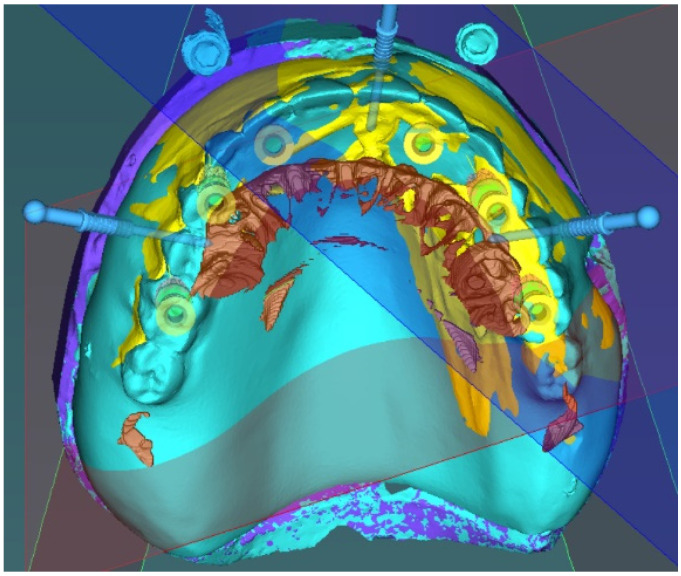
Registration of the optical scans and cone beam computed tomography (CBCT) reconstruction and the plan of the surgical template.

**Figure 2 materials-13-05759-f002:**
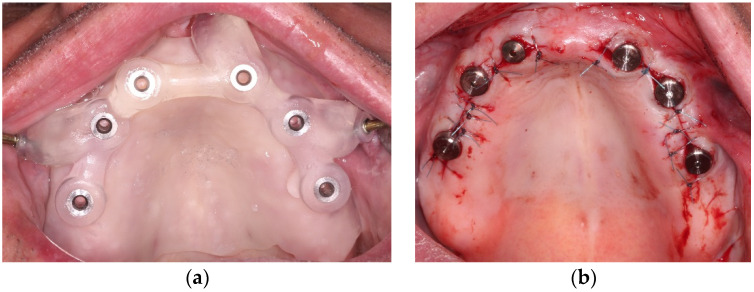
The surgical procedure: (**a**) The mucosa-supported, pin-anchored surgical template in place; (**b**) The implants placed in the edentulous upper jaw.

**Figure 3 materials-13-05759-f003:**
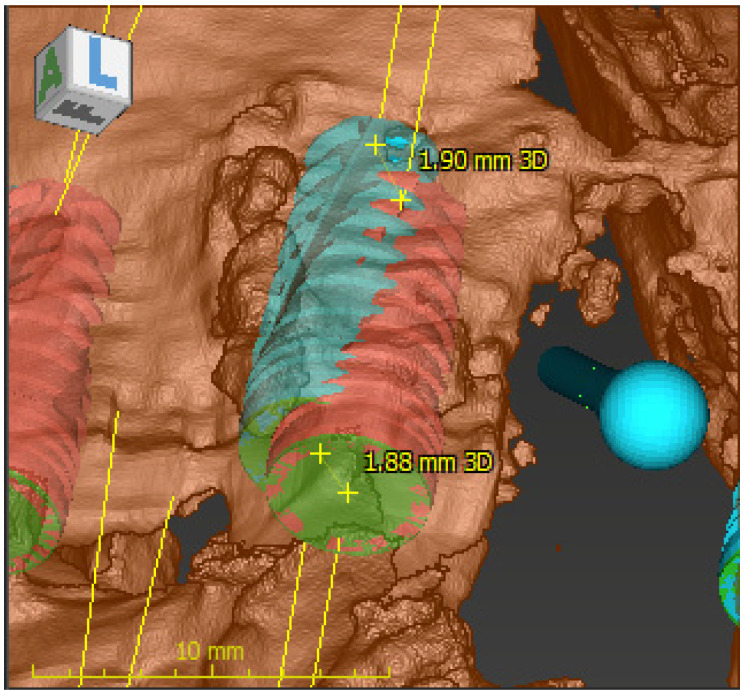
Measuring the primary outcome variables in the Treatment Evaluation plug-in of coDiagnostiX software, version 10.2 (Dental Wings, Montreal, CA, USA).

**Figure 4 materials-13-05759-f004:**
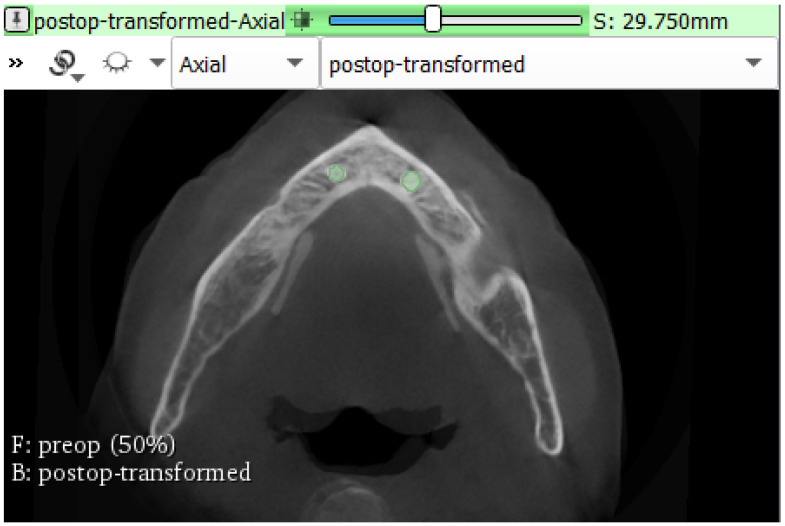
The calculation of the bone volume fraction (BV/TV) using the BoneTexture Extension of 3DSlicer 4.10.2 software (The Brigham and Women’s Hospital, Inc., Boston, MA, USA).

**Table 1 materials-13-05759-t001:** Descriptive and comparative statistics of the primary outcome variables between the two study groups.

Primary Outcome Variables		Test Group (Novice Surgeons)		Control Group (Experienced Surgeons)		
	Unit	M (Mean)	SD (Standard Deviation)	M (Mean)	SD (Standard Deviation)	*p*
Angular deviation ^1^	°	6.544	5.393	7.177	4.214	0.396
Coronal three-global deviation ^2^	mm	1.987	0.7049	1.879	0.7893	0.655
Apical global deviation ^2^	mm	1.954	0.6853	2.124	0.8373	0.477

^1^ Mann–Whitney’s U test was carried out to compare the angular deviation data of implants placed by the two study groups. ^2^ One-way ANOVA test was carried out to compare the coronal and apical global deviations of implants placed by the two study groups.

**Table 2 materials-13-05759-t002:** Correlation statistics between the primary outcome variables and bone density datasets.

Correlation between Parameters		R Value	Level of Significance (*p* Value)
Grey level measurements (HU)	Angular deviation	−0.331 *	0.037 *
	Coronal global deviation	−0.027	0.869
	Apical global deviation	−0.090	0.580
Bone volume fraction (BV/TV)	Angular deviation	−0.377 *	0.020 *
	Coronal global deviation	−0.049	0.771
	Apical global deviation	−0.189	0.255

Spearman’s test was carried out to analyze the correlations between grey level measurements in HU, BV/TV data and primary outcome variables. * *p* < 0.05.

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
