# Peer review of "The Influence of Surgical Experience and Bone Density on the Accuracy of Static Computer-Assisted Implant Surgery in Edentulous Jaws Using a Mucosa-Supported Surgical Template with a Half-Guided Implant Placement Protocol—A Randomized Clinical Study"

_materials, 2020, doi:10.3390/ma13245759_

Round 1

Reviewer 1 Report

I think the paper is good, well designed and well written, which may be of interest to the profession.

The introduction is well documented;the need for a new study was justified. The authors conclude the introduction by defining your objectives; however, you do not define a hypothesis, which is more specific than objectives and is amenable to explicit statistical evaluation.

The design of this clinical randomized study and its ethical aspects seem to me adequate. The authors explain the calculation of the sample size, eligibility criteria, pre-surgical and surgical procedure, definition of outcome variables and statistical procedure; nevertheless, you have not specified how the randomization was carried out.

Results are clearly presented and support the conclusions.

In my opinion, the discussion is correct and I am pleased to find a reference to an aspect, which could have been more controversial, i.e. the quantitative use of gray values in CBCT

Author Response

MK We would like to thank the reviewer for their time and effort to review the manuscript. The corrections suggested in the review report allows us to improve the manuscript so that its quality may reach the high standards of this esteemed Journal.

R1 I think the paper is good, well designed and well written, which may be of interest to the profession.

MK We would like to thank the reviewer for the commendation.

R1 The introduction is well documented;the need for a new study was justified. The authors conclude the introduction by defining your objectives; however, you do not define a hypothesis, which is more specific than objectives and is amenable to explicit statistical evaluation.

MK We agree that stating the null hypothesis is important for the statistical analysis and we have revised our manuscript in row 86-89:

Because of the controversy in the literature regarding the role of experience and bone density on the accuracy of CAIS our null hypotheses were that there is no difference in the accuracy of CAIS between experienced and novice implant surgeons and that there is no correlation between bone density and the accuracy of CAIS.

Row 317-321:

In the present study there was no significant difference in the primary outcome variables (angular deviation, coronal global deviation, and apical global deviation of the implants placed) between the two groups, which supported our null hypothesis; surgical experience did not seem to influence the accuracy of half-guided static-CAIS with a mucosa supported template anchored by pins in the edentulous jaw.

Row 353-361:

According to the results of the present study, there is a weak, statistically significant negative correlation between bone density (presented in both grey level measurements and Bone Volume Fraction) and angular deviation. Our null hypothesis was rejected. The higher the bone density at the sites of implant placement, the higher the accuracy of static-CAIS with a mucosa supported half-guided template anchored by pins. In the clinical practice numerous factors besides bone density may influence the accuracy of CAIS. Because of the weak correlation between bone density and the accuracy of CAIS reported in the literature [30,31] and in the present study, to control these risk factors further in vitro studies may be required to assess the role of bone density in the accuracy of CAIS.

R1 The design of this clinical randomized study and its ethical aspects seem to me adequate. The authors explain the calculation of the sample size, eligibility criteria, pre-surgical and surgical procedure, definition of outcome variables and statistical procedure; nevertheless, you have not specified how the randomization was carried out.

MK We have carried out a simple randomization using the random number generator function of Excel (Microsoft, Redmond, WA, United States). We have revised the manuscript in row 125-127: accordingly:

Simple randomization was carried out using the random number generator function of Excel (Microsoft, Redmond, WA, United States) to determine whether patients would be included in the test or control group.

R1 Results are clearly presented and support the conclusions. In my opinion, the discussion is correct and I am pleased to find a reference to an aspect, which could have been more controversial, i.e. the quantitative use of gray values in CBCT

MK We would like to thank the reviewer for the commendation and remarks. We agree that stating the null hypotheses and detailed description of the randomization process improves the quality of the manuscript. We hope that the reviewer finds the corrections made in the manuscript sufficient and recommends our paper for publishing in the Journal.

Reviewer 2 Report

paper needs revision for grammar and style

the objective of the study is not clear, 

the major issue is the methodology, the origins of the study sample is not clear, the study sample is very small(40 implants in 13 patients), the quality and volume of the available bone was not presented, very little information provided on static-CAIS, grey level measurements in Hounsfield 115 Units (HU), and Bone Volume Fraction (BV/TV),

why authors used the Pearson’s test and not the Spearman's

finally assessing the experience based on the subjective classification of clinicians based on the number of implants!! and finally, the results do not show any difference between groups and r values are all <0.5

Author Response

MK We would like to thank the reviewer for their time and effort to review the manuscript. The corrections suggested in the review report allows us to improve the manuscript so that its quality may reach the high standards of this esteemed Journal.

R2 paper needs revision for grammar and style

MK We have made a thorough revision on the grammar and style of the manuscript.

R2 the objective of the study is not clear, 

MK We have revised the purpose of the study in the end of the introduction section to clearly state the aim of the study and the null hypotheses to reject or to support in row 82-89:

The aim of our randomized controlled clinical study was to analyze the influence of surgical experience on the accuracy of static-CAIS in the edentulous jaws using a mucosa supported surgical template stabilized by pins with a half-guided implant placement protocol. The secondary aim of this study was to analyze the influence of bone density on the accuracy of guided implant placement. Because of the controversy in the literature regarding the role of experience and bone density on the accuracy of CAIS our null hypotheses were that there is no difference in the accuracy of CAIS between experienced and novice implant surgeons and that there is no correlation between bone density and the accuracy of CAIS.

R2 the major issue is the methodology, the origins of the study sample is not clear,

MK We have revised the inclusion criteria to explain the origin of the study sample in row 108-110:

Patients of the Department of Community Dentistry, Semmelweis University, presenting with an edentulous lower and/or upper jaw who were more than 18 years of age, and needed implant supported restorations were included in the present study

R2 the study sample is very small(40 implants in 13 patients),

MK Before enrolling patients in the study we have estimated the number of cases (implants) needed in each study group by power analysis by consulting an expert statistician and using the IBM SPSS Sample Power 3 software. We have found an in vitro (Cushen et al.) and a clinical study (Marei et al.) the results of which suggested that there is a significant difference between the accuracy of CAIS depending on surgical experience.

Cushen et al. found that angular deviation of implant placement is 2.60 ± 1.25 ° and 3.96 ± 1.64 ° in expert and novice groups respectively. According to the results of sample size calculation if α (false positive rate) was determined at 0.05 number of cases was 18-22 to achieve a power of 90-95%.

According to the results of Marei et al. buccolingual deviation was 3.7 ± 3.35 mm and 8.5 ± 6.3 mm in experienced and novice groups respectively. According to the results of sample size calculation if α (false positive rate) was determined at 0.05 number of cases was 12-14 to achieve a power of 90- 95%.

Consequently, we have decided to enrol 20 cases (number of implant placement) in each study group.

We have revised the manuscript in row 98-107 accordingly:

Patient sample size was based on previous in vitro and clinical studies [1816,2422,3432]. Cushen et al. found that angular deviation of implant placement is 2.60 ± 1.25 ° in the study group where  surgical interventions were carried out by experienced surgeons and 3.96 ± 1.64 ° in the study group where surgery was carried out by novice implant surgeons [18]. According to the results of sample size calculation if α (false positive rate) was determined at 0.05 number of cases was 18-22 to achieve a power of 90-95%. According to the results of Marei et al. buccolingual deviation was 3.7 ± 3.35 mm and 8.5 ± 6.3 mm in experienced and novice groups, respectively [24]. According to the results of sample size calculation if α (false positive rate) was determined at 0.05 number of cases was 12-14 to achieve a power  of 90-95%. Consequently approximately 20 cases (number of implant placement) were enrolled in each study group.

R2 the quality and volume of the available bone was not presented, very little information provided on static-CAIS, grey level measurements in Hounsfield 115 Units (HU), and Bone Volume Fraction (BV/TV),

MK We would like to thank the reviewer for this observation. Available bone quality and volume may be an important factor in the accuracy of CAIS. We have presented our methods for measurements made and the results in row 206-209:

To assess the bone volume at the sites of implant placement width and height of the alveolar ridge was measured in the coDiagnostiX software. Width of the ridges was measured at the platform level of the implants planned; ridge height was measured from the center of the implant platform along the axis of the planned implant position to the closest anatomical landmark.

and in row 242-249:

Grey level was 613.4 ± 304.7 HU (663.7 ± 294.5 HU and 576.8 ± 313.5 HU for novice and experienced groups respectively) and BV/TV was 0.6761 ± 0,3644 (0.7294 ± 0.339.2 and 0.637.3 ± 0.3847 for novice and experienced groups respectively) calculated at the sites of implant placement. Width of the alveolar ridge at the sites of implant placement was 7.688 ± 1.094 mm (7.567 ± 0.7300 mm and 7.786 ± 1.329 mm for novice and experienced groups respectively). Height of the alveolar ridge at the sites of implant placement was 15.89 ± 4.011 mm (16.02 ± 4.624 mm and 15.78 ± 3.542 mm for novice and experienced groups respectively).

R2 why authors used the Pearson’s test and not the Spearman's

MK We used the Pearson’s test to analyze the correlation between grey level measurements in HU and angular deviation because scatter plot suggested a linear correlation. However, Spearman’s test may be more appropriate in this case because by using the Pearson’s test we presume a linear correlation. We have revised the manuscript in row 235-236 accordingly:

Spearman’s test (two-tailed) was carried out to correlate angular deviation, coronal, and apical global deviation data to BV/TV and grey level data.

We have corrected the R- and p values in Table 2.

R2 finally assessing the experience based on the subjective classification of clinicians based on the number of implants!!

MK In our Department, the Department of Community Dentistry, where the present study was conducted only those professionals are allowed to start to train in implant surgery that have sufficient experience in more basic oral surgical procedures. That is why the novice implant surgeons participating in this study are well versed in the techniques of flap elevation, handling of the bone, and soft tissues, flap closure, etc. It is our observation that prosthetically correct positioning of the implants is the biggest weakness of the novice implant surgeon. In this aspect improvement comes with the number of implants placed. The authors of this manuscript understand that the number of implants placed alone does not always reflect proficiency in implant surgery, however, we believe that achieving the prosthetically correct implant position  -which is the outcome of this study- is easier for the surgeon that has placed more implants. However, we could not find a reference in the literature assessing the role of number of implants placed in surgical proficiency. That is why similarly to all previous studies on the influence of surgical experience on the accuracy of CAIS, we have classified surgical experience based on the number of implants placed prior to the study.

R2 and finally, the results do not show any difference between groups

MK Indeed, our results show that  there was no significant difference in the primary outcome variables (angular deviation, coronal global deviation, and apical global deviation of the implants placed) between the two groups, which supported our null hypothesis; surgical experience did not seem to influence the accuracy of half-guided static-CAIS with a mucosa supported template anchored by pins. In the literature there is a controversy about the influence of surgical experience on the accuracy of CAIS. However, to the best of our knowledge our manuscript is the first to assess the influence of surgical experience on the accuracy of half-guided CAIS with a mucosa supported template in the edentulous jaw.

R2 and r values are all <0.5

MK A correlation with an R-value below 0.5 is generally considered a weak correlation. However, there are a lot of factors influencing the accuracy of CAIS and bone density according to our study and some previous studies (Putra et al. and Ochi et al.) is only one of them. In the literature we were able to find only two previous article (Putra et al., Ochi et al.) where the influence of bone density on the accuracy of CAIS was assessed by correlating grey level values and outcome variables of accuracy in a homogenous sample. In their statistical analysis, Putra et al. found a statistically significant, weak correlation between mesiodistal deviation of implants placed with CAIS and bone density. Ochi et al. have found a statistically significant weak correlation between bone density and implant positioning depth following CAIS. Ozan et al. did not report R values in their study. That is why the authors of the present manuscript believe that although the correlation is weak the statistically significant negative correlation between bone density and angular deviation of CAIS is an important finding of our study. However, we recognise that the presentation of our results in the discussion may be misleading. That is why we have revised the manuscript in row 353-362 to tone down the interpretation of this finding:

According to the results of the present study, there is a weak, statistically significant negative correlation between bone density (presented in both grey level measurements and Bone Volume Fraction) and angular deviation, so our null hypothesis was rejected. The higher the bone density at the sites of implant placement, the higher the accuracy of static-CAIS with a mucosa supported half-guided template anchored by pins. In the clinical practice numerous factors besides bone density may influence the accuracy of CAIS. Because of the weak correlation between bone density and the accuracy of CAIS reported in the literature [30,31] and in the present study, to control these risk factors further in vitro studies may be required to assess the role of bone density in the accuracy of CAIS.

MK We would like to thank the reviewer for pointing out the shortcomings of the manuscript. The detailed explanation of the sample size calculation, detailed information on the quality and volume of the alveolar ridges, correction of the statistical analysis, and discussion of the weakness of the correlation between bone density and CAIS improves our manuscript. We hope that the changes made in the manuscript are sufficient and the reviewer recommends our paper for publishing in the Journal.

Reviewer 3 Report

The paper is a randomized clinical study on the effectiveness of a surgical guide supported by mucosa in providing a support to the clinician during implantology.

The authors made a great work in terms of methodology and the paper sounds scientific and well written.

However some improvements are mandatory before acceptance.

  • Too many double spaces were found during the revision: please fix it.
  • Line 23: too many ands… “Angular deviation, coronal, and apical global deviation, and grey level…”: I suggest to remove the first one.
  • In the introduction, emphasis is given to “The number, dimensions, position, and inclination of dental implants placed are determined by the prosthetic planning, which facilitates optimal esthetic results, function, ideal biomechanical loading for all components of the prosthesis, and long term stability of soft- and hard tissues surrounding  the  ” But the importance of surface treatment of the fixture should be also underlined, since it could modify the mucosal aspect around the implant, giving different outcome when compared to different treated surfaces. I suggest the following articles to enrich the introduction about this aspect:
  1. “Guarnieri R, Di Nardo D, Gaimari G, Miccoli G, Testarelli L. Short vs. Standard Laser-Microgrooved Implants Supporting Single and Splinted Crowns: A Prospective Study with 3 Years Follow-Up. J Prosthodont. 2019 Feb;28(2):e771-e779.”
  2. “Lollobrigida M, Fortunato L, Serafini G, Mazzucchi G, Bozzuto G, Molinari A, Serra E, Menchini F, Vozza I, De Biase A. The Prevention of Implant Surface Alterations in the Treatment of Peri-Implantitis: Comparison of Three Different Mechanical and Physical Treatments. Int J Environ Res Public Health. 2020 Apr 11;17(8):2624.”
  • Figure 1: please separate “theplan”.
  • Materials and methods are clear and well explained. Different aspects are analyzed with a dedicated statistical test. The authors did a great job in the explication of all the variables identified and included in the study.
  • Results are easy to understand and comprehensive. All the studied characteristics were reported in tables which are clear and concise.
  • Discussion: this section is complete and evaluates the outcome of different papers present in literature. The overall is comprehensive, concise and complete in its various aspects.
  • Conclusions are concise and clear.
  • Bibliography is formatted respecting the journal’s requirements and no improper citations are evidenced.
  • Figures and labels are clear and easy to comprehend.

English is clear and easy to understand.

Author Response

MK We would like to thank the reviewer for their time and effort to review the manuscript. The corrections suggested in the review report allows us to improve the manuscript so that its quality may reach the high standards of this esteemed Journal.

R3 The paper is a randomized clinical study on the effectiveness of a surgical guide supported by mucosa in providing a support to the clinician during implantology. The authors made a great work in terms of methodology and the paper sounds scientific and well written.

MK We would like to thank the reviewer for the commendation.

R3 However some improvements are mandatory before acceptance. Too many double spaces were found during the revision: please fix it.

MK We have found double spaces in rows 14, 61, 108, 165, 348, and 350 and we have corrected them.

R3 Line 23: too many ands… “Angular deviation, coronal, and apical global deviation, and grey level…”: I suggest to remove the first one.

MK We have removed the redundant “and”.

R3 In the introduction, emphasis is given to “The number, dimensions, position, and inclination of dental implants placed are determined by the prosthetic planning, which facilitates optimal esthetic results, function, ideal biomechanical loading for all components of the prosthesis, and long term stability of soft- and hard tissues surrounding  the  ” But the importance of surface treatment of the fixture should be also underlined, since it could modify the mucosal aspect around the implant, giving different outcome when compared to different treated surfaces. I suggest the following articles to enrich the introduction about this aspect:

  1. “Guarnieri R, Di Nardo D, Gaimari G, Miccoli G, Testarelli L. Short vs. Standard Laser-Microgrooved Implants Supporting Single and Splinted Crowns: A Prospective Study with 3 Years Follow-Up. J Prosthodont. 2019 Feb;28(2):e771-e779.”
  2. “Lollobrigida M, Fortunato L, Serafini G, Mazzucchi G, Bozzuto G, Molinari A, Serra E, Menchini F, Vozza I, De Biase A. The Prevention of Implant Surface Alterations in the Treatment of Peri-Implantitis: Comparison of Three Different Mechanical and Physical Treatments. Int J Environ Res Public Health. 2020 Apr 11;17(8):2624.”

MK Surface modification of implants and fixtures during production and treatment of peri-implantitis is indeed an important aspect of long-term success in rehabilitation with dental implants. We have made a revision in row 41-43 accordingly citing the studies suggested by the reviewer.

Surface treatment of dental implants and fixtures during production plays an important role in muco-integration. Treatment modalities for peri-implantitis should be chosen with regard to the fact that they may alter surface physicochemical properties [4,5].

R3 Materials and methods are clear and well explained. Different aspects are analyzed with a dedicated statistical test. The authors did a great job in the explication of all the variables identified and included in the study.

MK We would like to thank the reviewer for the commendation on the materials and method section.

R3 Figure 1: please separate “theplan”.

MK We have corrected the legend for Figure 1.

R3 Results are easy to understand and comprehensive. All the studied characteristics were reported in tables which are clear and concise.

R3 Discussion: this section is complete and evaluates the outcome of different papers present in literature. The overall is comprehensive, concise and complete in its various aspects.

R3 Conclusions are concise and clear.

R3 Bibliography is formatted respecting the journal’s requirements and no improper citations are evidenced.

R3 Figures and labels are clear and easy to comprehend.

R3 English is clear and easy to understand.

MK We would like to thank the reviewer for the commendation on the manuscript. We would like to thank the reviewer for the correction of grammatical errors made and the suggestions about the introduction. We believe that our manuscript benefits from the broader background information. We hope that the reviewer finds the corrections made in the manuscript sufficient and recommends our paper for publishing in the Journal.

Round 2

Reviewer 2 Report

thank you for the revisions, I recommend revising the literature review and focusing on the topic of the study, paper still benefits from language editing 

Author Response

R2 thank you for the revisions,

MK We would like to thank your efforts to point out the errors in the manuscript!

R2 I recommend revising the literature review and focusing on the topic of the study.

MK In row 41-44 the background information on the muco-integration and surface modification of implant fixtures was specifically requested by another reviewer to elaborate on soft tissue stability.

R2 paper still benefits from language editing 

MK The manuscript has been edited by the MDPI language services to assess this problem.

We would like to thank the reviewer once more for their work and we hope that the reviewer finds the changes made sufficient to recommend our manuscript for publication.